# Formation of a Double-Layer Ultrafine Crystal Structure for High-Current Pulsed Electron Beam-Treated Al–20Si–5Mg Alloy

**Bo Gao \*, Kui Li and Pengfei Xing**

School of Metallurgy, Northeastern University, Shenyang 110004, China
* Correspondence: gaob@smm.neu.edu.cn; Tel.: +86-24-8368-1320

**Abstract:** In this paper, the effect of high-current pulsed electron beam (HCPEB) on the microstructure refinement of an Al–20Si–5Mg alloy in the cross-section modified zone was studied, and a double-layer ultrafine crystal structure of the Al–20Si–5Mg alloy was formed. It was found that the cross-section modified zone was divided into three zones, namely, the remelted layer, the heat-affected zone, and the thermal stress wave-affected zone after HCPEB treatment. For the remelted layer, metastable structures were formed due to the rapid heating and cooling rates. For the heat-affected zone, the grain of the aluminum phase was refined due to the cooperative effects of shock wave (formed during an eruption event of the brittle phase), thermal-stress wave (formed during thermal expansion of the alloy surface), and quasi-static thermal stress (formed as a result of an unevenly distributed temperature gradient in the inner material) at high temperatures. For the thermal stress wave-affected zone, the grain refinement was not obvious due to the decreasing energy of the shock wave and the thermal-stress wave at low temperatures. In addition, firm evidence for the tracing of shock waves in the heat-affected zone was demonstrated for the first time and verified for the founding of the broken acicular eutectic silicon. Through this experiment, the mechanical properties of Al–20Si–5Mg alloy materials in both the remelted layer and heat-affected zone were significantly improved after HCPEB treatment.

**Keywords:** high-current pulsed electron beam; heat-affected zone; shock wave; thermal stress wave; grain refinement

## 1. Introduction

As an advanced surface modification technique, high-current pulsed electron beam (HCPEB) technology has attracted scholarly attention in the field of material surface modification owing to the advantages of this technique, including slight deformation of the workpiece, energy conservation, high efficiency, flexible processing method, and good repeatability [1–4].

Proskurovsky et al. [5] proposed the concept of "deep modification" for the first time, in which HCPEB modified not only the surface layer but also the deep structures of materials. Some researchers proposed that the cross-section of the material after HCPEB treatment be divided into three zones: the remelted layer (a few microns), the heat-affected zone (dozens of micrometers), and the thermal stress wave-affected zone (hundreds of microns) [5–8]. Qin et al. [9] studied the effect of HCPEB treatment on the microstructure of 45# steel and found two corrugated lines parallel to the boundary of one grain located close to 0.5 mm below the surface. They considered that these phenomena were ascribed to the effect of a stress wave. Moreover, some researchers found that microstructures in the deep zone were modified and that the mechanical properties also improved [9,10]. However, there is no clear evidence that demonstrates the tracing of the refinement in the materials. During the HCPEB treatment

process, temperature-induced dynamic thermal stress fields can generate thermal stress and shock stress waves. A thermal stress wave has small amplitudes of less than 0.1 MPa. A shock stress wave, however, is an atypical nonlinear wave, several hundreds of MPa in amplitude and much stronger than a thermal stress wave. Hence, it has a strong impact on material structure and properties far beyond the heat-affected zone [9].

In this work, HCPEB technology was used to treat the surface of Al–20Si–5Mg alloys. The tracing of the refinement was further studied, and the deep modification effect of HCPEB in the heat-affected zone was also studied in detail and verified.

## 2. Experimental

### 2.1. Sample Preparation

In this experiment, the chemical composition of the hypereutectic Al–Si–Mg alloy was as follows: 20 wt.% Si, 5 wt.% Mg, and 75 wt.% Al. The preparation process of the alloy before HCPEB treatment was as follows: First, pure aluminum (99.79 wt.% Al), industrial silicon of type 2202 (Si), and pure magnesium (99.9 wt.% Mg) were cut into small pieces and mixed according to specific proportions. The raw materials were dried, heated, and kept at 780 °C for 3 h and then cast in a 304 stainless steel mold with dimensions of $\Phi\,15 \times 80$ mm$^2$ at 730 °C. Next, the alloy ingots were cut into $10 \times 10 \times 10$ mm$^3$ cubes by electrical discharge machining and then polished using different diamond sandpapers (100#, 240#, 400#, 800#, 1500#, 3000#) and 1 μm diamond abrasive paste. Finally, the surfaces of the samples were cleaned with anhydrous ethanol and dried quickly.

### 2.2. HCPEB Treatment

For surface modification of the samples, HCPEB treatment was carried out with MMLAB-HOPE-I HCPEB equipment. The corresponding process parameters are shown in Table 1. Considering the accumulating effect of HCPEB energy deposition and our previous experimental results [6], the numbers of pulses were set to 5, 15, and 25. Further details on the operating principle of the HCPEB system can be found in [11].

**Table 1.** Process parameters of high-current pulsed electron beam.

| Background Vacuum ($10^{-3}$ Pa) | Acceleration Voltage (kV) | Energy Density (J/cm$^2$) | Distance from Anode to Target (cm) | Pulse Duration (μs) | Pulse Interval (s) |
|---|---|---|---|---|---|
| 6 | 27 | 5–6 | 10 | 2 | 10 |

### 2.3. Characterization

In this experiment, the morphology of the hypereutectic Al–20Si–5Mg alloy in terms of the surface and cross-section before and after HCPEB treatment was examined by means of field emission scanning electron microscopy (FESEM Hitachi S-4800, Tokyo, Japan) with energy-dispersive X-ray spectroscopy (EDS) and electron backscattering diffraction (EBSD) attachments. The phase analysis of the sample surface was conducted using a Panalytical X'Pert PRO PW3040/60 X-ray diffractometer (XRD, Almelo, The Netherlands) with a Cu K$\alpha$ ($\lambda = 0.154$ nm). Metastable structures after HCPEB treatment were observed through transmission electron microscopy (TEM, Tecnai G20, Hillsboro, OR, USA) after being electro-polished in a twin-jet machine with 6% perchloric acid alcoholic solution at about −20 °C. The morphology of the alloy at the cross-section after HCPEB treatment was examined by EBSD after vibration polishing with a VibroMet 2 machine (Buehler, Lake Bluff, IL, USA) with 0.04 m SiO$_2$ polishing solution for 6 h. Beam control mode was applied for automatic orientation mapping with a step size of 150 nm. Moreover, a nanoindentation test was conducted using a Hysitron TI 980 Triboindenter instrument (Bruker, Billerica, MA, USA) under 5-10-5 loading mode (loading for 5 s, pause for 10 s, unloading for 5 s) with a load of 2 mg. Finally, the selected zone subjected to SEM and

TEM analyses was first observed in low power and a local zone that represented the entire specimen was selected in the high power and the SEM and TEM images were finally obtained.

## 3. Results and Discussion

### 3.1. SEM Analysis of the Surface Morphology

Figure 1 shows SEM images of the Al–20Si–5Mg alloy surface microstructures before and after HCPEB treatment. As seen from Figure 1a, the microstructures of the Al–20Si–5Mg alloy were mainly composed of the acicular eutectic silicon phase, the plate-like $Mg_2Si$ phase, the primary silicon phase, and the aluminum matrix before HCPEB treatment, which is consistent with [12]. As shown in Figure 1b–d, many crater structures formed on the HCPEB-treated surface due to the eruption of the local structures (Si and $Mg_2Si$) in the sample's subsurface. During the HCPEB process, the structures on the surface of the alloy can rapidly melt and solidify, however, local structures still remain in a melted state (due to the lower thermal conductivity of Si (150 W/(m·K)) and $Mg_2Si$ (3 W/(m·K)) compared to Al (237 W/(m·K)), heat accumulation in the center of coarse Si and $Mg_2Si$ phases, and the temperature of these phases reaches a much higher value than that of the surrounding $\alpha(Al)$ phase in very short time (several $\mu s$), which is apt to erupt to reduce temperature, reach heat balance, and form the shock wave [9]. In addition, Figure 1b–d also shows that the crater density decreased gradually with the increase in pulse numbers. This phenomenon is attributed to the polishing effect during HCPEB treatment [13].

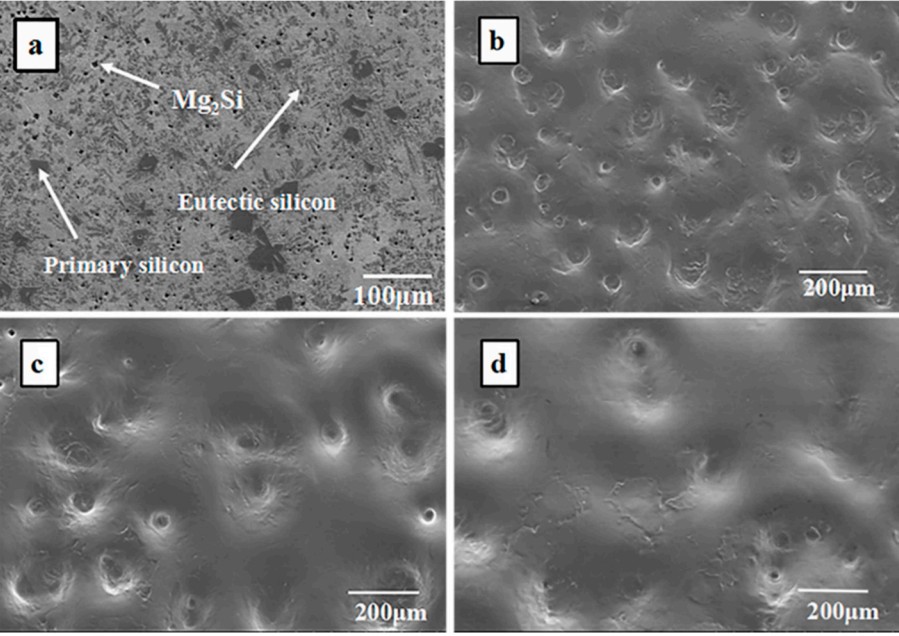

**Figure 1.** Scanning electron microscopy (SEM) morphology images of Al–20Si–5Mg alloy before and after high-current pulsed electron beam (HCPEB) treatment: (**a**) Initial sample, low magnification; (**b**) 5 pulses, high magnification; (**c**) 15 pulses, high magnification; and (**d**) 25 pulses, high magnification.

Figure 2 shows the distribution of elements on the surface after HCPEB treatment with 25 pulses. As seen from Figure 2b–d, the Al matrix and crater structures combined very well, and the distribution of Al was more uniform on the top alloy surface after HCPEB treatment due to the effect of element diffusion [14]. However, Mg and Si elements aggregated at the bottom of the crater structures due to the low diffusion rate of Mg and Si. Only small amounts of Mg and Si elements diffused into the Al matrix due to the enhanced element diffusion by the HCPEB treatment [14].

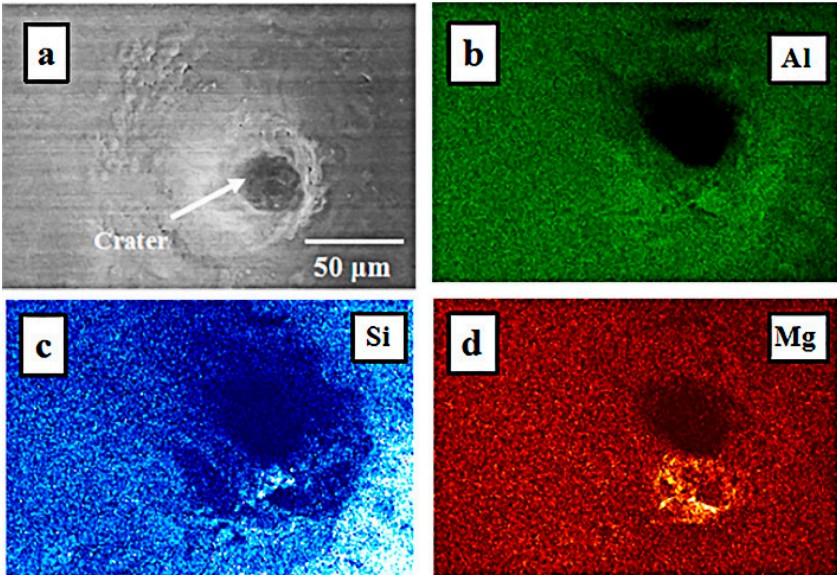

**Figure 2.** Element distribution maps of Al–20Si–5Mg alloy surface after HCPEB treatment with 25 pulses: (**a**) BSE (Back scattered electron) image; (**b**) element distribution of Al; (**c**) element distribution of Si; and (**d**) element distribution of Mg.

*3.2. TEM Analysis of the Surface Morphology*

Figure 3 shows the TEM image of the metastable structures of the Al–20Si–5Mg alloy surface after HCPEB treatment with 25 pulses. Figure 3a shows that nano-primary silicon phases with grain sizes of 100–200 nm were dispersed in the Al matrix, and the nano-crystallites were identified as Si by the corresponding selected area electron diffraction (SAED). The nano-primary silicon phase was verified as part of the crater structure (primary silicon) in our previous work [15]. The formation mechanism of the nano-primary silicon phase can be explained as follows: (1) The alloy surface melts rapidly due to the rapid heating rate of HCPEB; (2) the atoms of liquid silicon continuously migrate to the crystal nuclei of silicon and grow into silicon grains; (3) the silicon grain solidifies rapidly without further growth due to the rapid cooling rate. Hence, nano-primary silicon is formed.

Figure 3b indicates that many micro nano-eutectic silicon phases (5–20 nm) formed near the edge of the crater structures. This phenomenon confirms the findings in [16]. The nano-crystallites were identified as Si by the corresponding SAED pattern. The formation of these nano-eutectic silicon phases is also attributed to the rapid heating and cooling rates of the HCPEB treatment. Some eutectic silicon phases were transformed into a supersaturated solid solution in the aluminum matrix, and the remaining silicon atoms aggregated at the grain and sub-grain boundaries of the Al phase, thus forming nano-eutectic silicon phases.

Figure 3c shows the nano-$Mg_2Si$ phase with a grain size of 5–20 nm, which was formed and dispersed uniformly on the alloy surface after the HCPEB treatment. The corresponding SAED analysis demonstrates the polycrystalline as the $Mg_2Si$ phase. The nano-$Mg_2Si$ phase can be used for pinning of the grain and sub-grain boundaries, which improves the deformation resistance and surface strength of the material.

Figure 3d shows the formation of nano-Al cellular structures with a size of 100–200 nm, induced after the HCPEB treatment. In addition, many micro-sized grey particles were distributed at the grain and sub-grain boundaries of the Al phase. These microparticles were verified to be nano-silicon by the corresponding SAED pattern. The authors of [14] reported that these fine nano-precipitates were formed by the diffusion of Si atoms to the grain and sub-grain boundaries of the Al phase during the HCPEB treatment process.

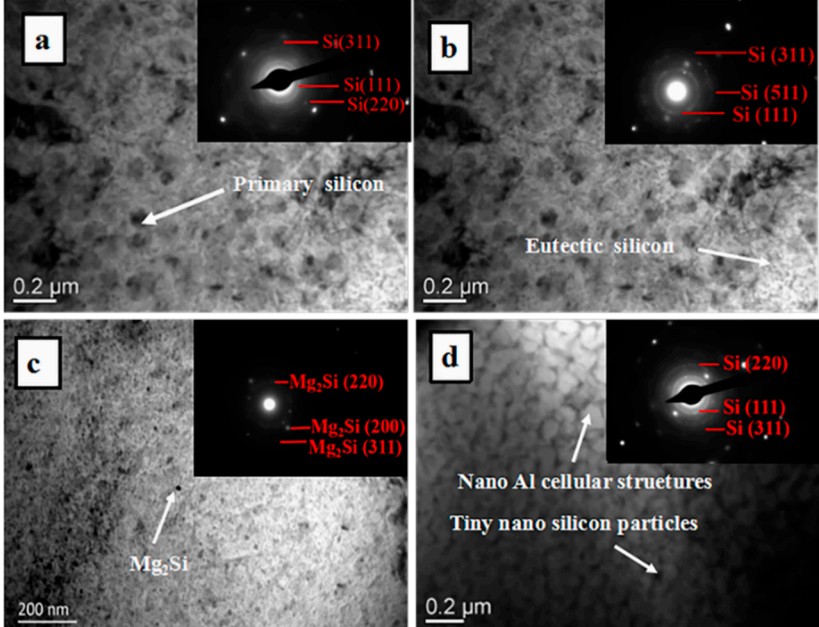

**Figure 3.** The formation of metastable structures of Al–20Si–5Mg alloy after HCPEB treatment with 25 pulses: (**a**) nano-primary silicon phase and the corresponding selected area electron diffraction (SAED) pattern; (**b**) nano-eutectic silicon phases and the corresponding SAED pattern; (**c**) nano-Mg$_2$Si phase and the corresponding SAED pattern; and (**d**) nano Al cellular structures, tiny nano silicon particles, and the corresponding SAED pattern.

In conclusion, these metastable structures were formed after HCPEB treatment. Meanwhile, these structures significantly improved the surface properties of the materials [6,13]. Therefore, HCPEB is one of the most effective methods for preparing metastable materials.

### 3.3. XRD Analysis of the Surface Morphology

Figure 4 shows the XRD diffractograms of the Al–20Si–5Mg alloy before and after HCPEB treatment. As Figure 4a indicates, no new phase was formed after HCPEB treatment, and the sample before and after treatment was mainly composed of three phases: Si, Mg$_2$Si, and Al phases. Figure 4b shows that the diffraction peaks of Al, Si, and Mg$_2$Si presented a clear broadening phenomenon. This phenomenon became more obvious with increasing the pulse number, which may be attributed to the refinement of grains, the existence of structural defects, and the effect of the stress state [17]. This phenomenon is consistent with the results of the TEM analysis.

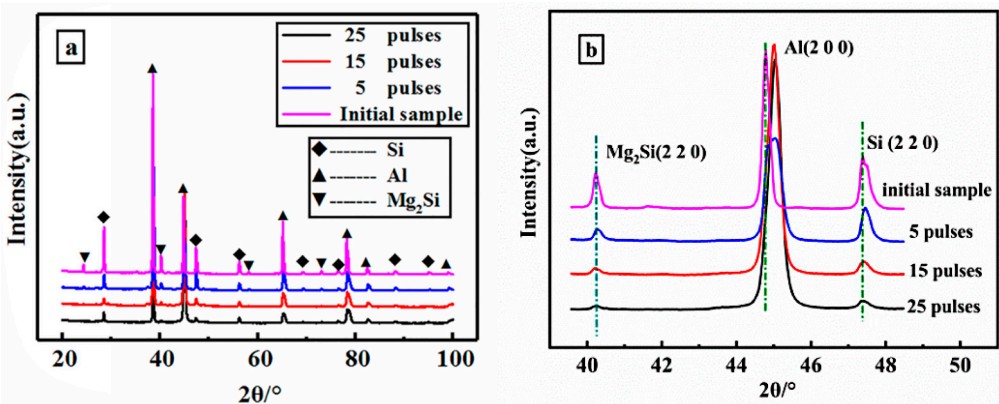

**Figure 4.** The X-ray diffractometer (XRD) diffractograms of Al–20Si–5Mg alloy surface before and after HCPEB treatment: (**a**) XRD diffractograms; (**b**) local enlargement of (**a**).

### 3.4. SEM and EBSD Analyses of the Cross-Sectional Morphology

Figure 5 shows that the eutectic silicon phases (approximately 30 μm beneath the top surface layer) were broken down along the HCPEB-irradiated direction in the heat-affected zone after HCPEB treatment with 25 pulses. The eutectic silicon was broken down into two segments, demonstrating that the HCPEB technology could modify not only the structures of the top melted surface but also those in the heat-affected zone.

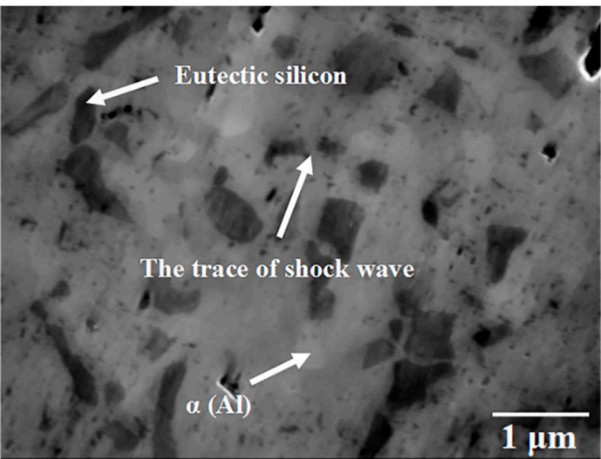

**Figure 5.** SEM morphology of thermal-affected zone in cross-sectional microstructures of Al–20Si–5Mg alloy after HCPEB treatment with 25 pulses.

During the HCPEB treatment process, if the HCPEB energy is in the melting mode, the energy of the pulse is extremely small, and only the surface of the material melts. Then the alloy surface will undergo rapid melting and cooling processes, leading to the formation of a thermal stress wave in which the wave energy is weak, the quasi-static thermal stress has a direction perpendicular to the HCPEB-irradiated direction, and it only exists in the heat-affected zone [9]. If the HCPEB energy is in vaporization mode, the energy of the pulse is extremely high and the vaporization of the material surface occurs. This will lead to strong vaporization and eruption in the local brittle phase zone of the subsurface, resulting in the formation of a new shock wave in addition to the thermal stress wave and quasi-static thermal stress. The newly formed shock wave will be relatively high and propagate inside the material [18].

In this experiment, the energy density of the HCPEB treatment was 5–6 J/cm$^2$, which led to strong vaporization and the eruption of the Al–20Si–5Mg alloy surface. The heat of the alloy surface spread to the inner material, leading to a high temperature of the structures in the heat-affected zone. Meanwhile, a high-energy shock wave penetrated the eutectic silicon phase, and the brittle eutectic silicon phases were broken down in the heat-affected zone. The EBSD analysis of the alloy cross-section after 25 pulses is shown in Figures 6 and 7. Figure 6 shows that the phases of Al, Mg$_2$Si, and Si were refined in the heat-affected zone. The grain size was measured by an EBSD soft tool, and as the corresponding grain size distribution in the heat-affected zone (Figure 7) indicates, most of the Al phases were less than 1 μm, which demonstrates a clear grain refinement effect of the HCPEB treatment. Figure 6 also shows that a large-sized Al phase appeared around the thermal stress wave-affected zone, indicating that the grain refinement effect was not obvious in this zone since the influences of the shock wave and the thermal stress wave decreased gradually with increasing depth. Furthermore, the figure shows that the quasi-static thermal stress formed only in the heat-affected zone [9]. Furthermore, the structure was constantly in a homoeothermic state during the entire HCPEB treatment process, and the grain was therefore difficult to refine.

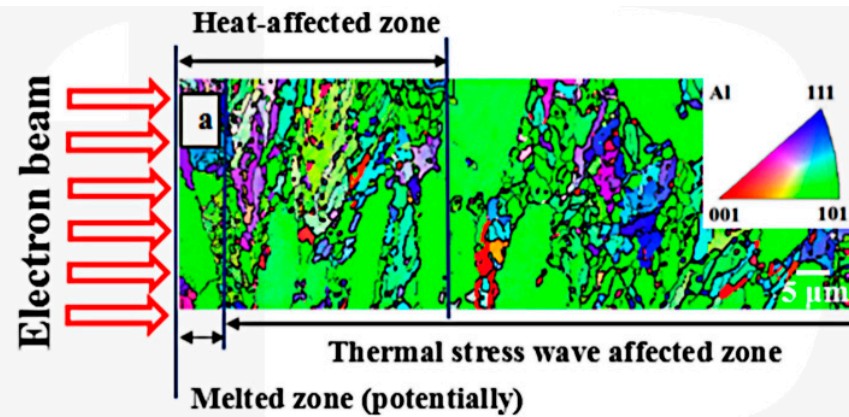

**Figure 6.** The cross-sectional electron backscattering diffraction (EBSD) orientation image after 25 pulse treatments.

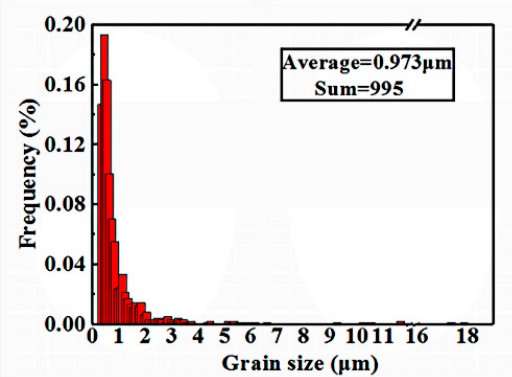

**Figure 7.** Grain size distribution of Al in the cross-sectional thermal-affected zone after 25 pulse treatments.

Grain refinement in the heat-affected zone is attributed to the following three aspects: (1) The material is in a high-temperature state during the entire HCPEB treatment process, so the microstructures of the materials can be easily refined. (2) The thermal stress wave and quasi-static thermal stress lead to the formation of grain boundary slip (GBS) and the transformation of the shearing surface into shearing bands of different orientations. The grain boundary sliding at the original grain boundary results in the generation of a strain gradient and rotation of the grains along the deformation zone at a large angle. The shearing bands lead to a continuous breaking of coarse grains into small grains and a rapid increase in the amount of boundary orientation differences for the small grains [19,20]. (3) A shock wave propagates and thus breaks the coarse grains into small pieces. This series of processes eventually leads to the formation of new fine grains in the heat-affected zone.

In brief, the TEM analysis results show that nano-primary silicon, eutectic silicon, $Mg_2Si$, and aluminum cellular structures formed in the remelted layer after HCPEB treatment. The EBSD and SEM results show that sub-micron aluminum, $Mg_2Si$, and eutectic silicon phases formed in the heat-affected zone. The formation mechanism of double-layer structures (the remelted layer and the heat-affected zone) can be summarized as follows: The former results are ascribed to the rapid heating and cooling rates of the HCPEB treatment and the latter results are attributed to the cooperative effect of the shock wave, thermal stress wave, and quasi-static thermal stress under the high-temperature condition of the HCPEB treatment.

### 3.5. Nanoindentation Analysis of the Cross-Sectional Structure

Figure 8 shows the results of the nanoindentation test for the Al phase of the HCPEB-treated cross-section of Al–20Si–5Mg alloy. Apparently, the nano-hardness distribution in the HCPEB-treated cross-section of Al–20Si–5Mg alloy can be divided into three zones, as shown in Figure 8a: the remelted layer, the heat-affected zone, and the thermal stress wave-affected zone. The nano-hardness of the aluminum phase in the remelted layer was higher than that in the heat-affected and thermal stress wave-affected zones. The nano-hardness value decreased with an increase in the depth from the remelted layer to the thermal stress wave-affected zone. In addition, the nano-hardness fluctuated slightly in the heat-affected zone, and this fluctuation is attributed to the dislocation multiplication in the deep layer of the material at a certain degree. The dislocation multiplication is prone to result in the formation of jogs through the interaction of dislocations, further inducing dislocation tangles. As a result, dislocation motion was seriously inhibited, resulting in hard plastic deformation and thus improving the strength of the material. Therefore, the fluctuation of the nano-hardness distribution shown in Figure 8a forms in the cross-section [1,21,22]. Figure 8b shows load-depth curve charts (the depth of maximum indentation and the depth of unloading) of the Al phase from the remelted layer to the thermal stress wave-affected zone. It can be seen that the depth of the remelted layer was shallower than that of the heat-affected zone and the thermal stress wave-affected zone, indicating a larger nano-hardness of the Al phase in the remelted layer, which is attributed to the refinement of grains based on the Hall–Petch formula [23]. The grain refinement led to an increase in the amount of grain boundaries. Since grain boundaries can hinder the slippage of dislocations, dislocation was seriously inhibited at the grain boundaries, further affecting the slippage motion of subsequent dislocations. Thus, the hardness of the Al–Si alloy was improved. The increase in the amount of grains can also benefit the cooperative deformation between grains during plastic deformation, further improving the plasticity of the alloy [24].

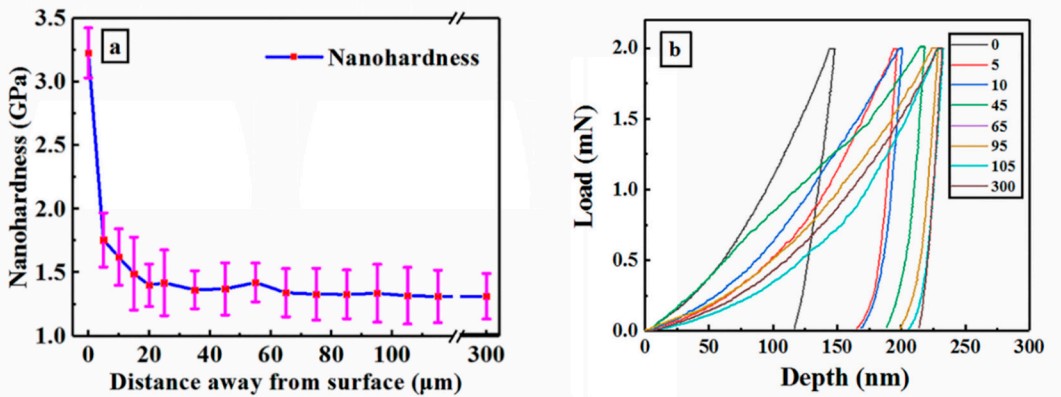

**Figure 8.** The cross-sectional nanoindentation tester results of Al–20Si–5Mg alloy after HCPEB treatment with 25 pulses: (**a**) evolution of nanohardness with distance away from surface; (**b**) evolution of load with depth away from top surface.

In summary, for the remelted layer, the improved hardness of the Al phase was the result of grain refinement, which occurred due to the rapid melting and cooling rates of the HCPEB treatment. The improved nano-hardness in the heat-affected zone is attributed to the grain refinement due to the cooperative effects of the shock wave, the thermal stress wave, and the quasi-static thermal stress under high-temperature conditions of the HCPEB treatment.

## 4. Conclusions

- Ultrafine grains with double layers (remelted layer + heat-affected zone) in the Al–20Si–5Mg alloy are formed by the HCPEB treatment. For the remelted layer, nano-primary silicon, eutectic silicon, $Mg_2Si$, and aluminum cellular structures are generated after the HCPEB treatment owing to the rapid heating and cooling rates. For the heat-affected zone, sub-micron Al, $Mg_2Si$, and eutectic silicon phases are formed due to the cooperative effects of shock waves, thermal stress waves, and quasi-static thermal stress under high-temperature conditions.

- HCPEB can improve the mechanical properties of materials in not only the remelted layer but also the heat-affected zone. The improvement in hardness of the aluminum phase in the remelted layer is the result of grain refinement, which arises due to the rapid melting and cooling rates of the HCPEB treatment. The improvement of nano-hardness in the heat-affected zone is attributed to grain refinement due to the cooperative effect of the shock wave, thermal stress wave, and quasi-static thermal stress under high-temperature conditions.

- Solid evidence for the tracing of the shock wave in the heat-affected zone was demonstrated for the first time and was verified for the formation of broken acicular eutectic silicon in the Al–20Si–5Mg alloy, demonstrating the deep modification effect of shock waves on the cross-section microstructures.

**Author Contributions:** Methodology, B.G.; software, P.X.; investigation, P.X.; data curation, K.L.; writing—original draft preparation, K.L.; project administration, B.G.

**Funding:** This study is supported by the National Natural Science Foundation of China (51671052) and the Fundamental Research Funds for the Central Universities (N182502042).

**Acknowledgments:** B.G. would like to thank N. Xu's discussion about EBSD analysis.

**Conflicts of Interest:** The authors declare no conflict of interest.

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
