# Peer review of "Formation of a Double-Layer Ultrafine Crystal Structure for High-Current Pulsed Electron Beam-Treated Al–20Si–5Mg Alloy"

_coatings, doi:10.3390/coatings9070413_

Round 1

Reviewer 1 Report

The authors present an interesting investigation on the effect of high-current pulsed electron beam (HCPEB) on the micro-structure refinement of an Al-20Si-5Mg alloy, showing details about the resulting structures along the cross-section of the material and the corresponding properties in terms of nano-hardness. The reviewer suggests to accept the article after minor revisions:

lines 61-66: the reviewer suggests to collect the process parameters in a table. Moreover, the authors should improve the explanation of the choice of the corresponding values and in particular about the number of pulses.

lines 90-92: improve the English of the sentence. It is not clear what causes the eruption and the subsequent formation of the shock wave.

line 149: the authors state that the metastable structures improve the surface properties. The reviewer suggests to add some more information about how they partecipate to such an improvement supporting with some references.

lines-173-180: avoid using many brackets. The reader tends to lose the thread of the sentence.

Reviewer 2 Report

This is an interesting piece of research. However, there are some shortcomings which need to be addressed before it is suitable for publication.

First, the Authors should consider restricting the speculative part of the paper.  If they write that some features are "due to the effect of element diffusion" should  provide proof of that or concentrate the attention on the feature as such. Another example, if they write "TEM images of the metastable structures.." they should explain why their metastable, or simply delete the term "metastable". Can they provide arguments why the formation mechanism of Mg2Si is similar to that of the nano-eutectic ....?

There is also a major problem with the use of the term "thermal wave". Intuitively it might be understand as the related to the time dependent profile of the temperature as a function of the distance from the coated surface. No formal definition is offered. Also, no estimate is provided of amplitude and time characteristics of the "heat wave".

Explanation should be offered how the "shock wave ... breaks the coarse grains into small pieces".

The way of measuring the grain size is not explained.

The way the SEM and TEM images have been obtained should be explained in more details, in particular in the context of the images presented being representative for the entire specimen. 

Reviewer 3 Report

Dear authors, you did great work. I recommend accepting this article in the present form.

Author Response

Dear authors, you did great work. I recommend accepting this article in the present form.

Answer: Thank you for your comments.

Round 2

Reviewer 2 Report

The manuscript has been improved and can be accepted for publication.